# Robust and Explainable Depression Identification from Speech Using Vowel-Based Ensemble Learning Approaches

Kexin Feng
*Department of Computer Science and Engineering*
*Texas A&M University*
College Station, Texas, USA
kexin@tamu.edu

Theodora Chaspari
*Institute of Cognitive Science, Department of Computer Science*
*University of Colorado Boulder*
Boulder, Colorado, USA
theodora.chaspari@colorado.edu

*Abstract*—This study investigates explainable machine learning algorithms for identifying depression from speech. Grounded in evidence from speech production that depression affects motor control and vowel generation, pre-trained vowel-based embeddings, that integrate semantically meaningful linguistic units, are used. Following that, an ensemble learning approach decomposes the problem into constituent parts characterized by specific depression symptoms and severity levels. Two methods are explored: a "bottom-up" approach with 8 models predicting individual Patient Health Questionnaire-8 (PHQ-8) item scores, and a "top-down" approach using a Mixture of Experts (MoE) with a router module for assessing depression severity. Both methods depict performance comparable to state-of-the-art baselines, demonstrating robustness and reduced susceptibility to dataset mean/median values. System explainability benefits are discussed highlighting their potential to assist clinicians in depression diagnosis and screening.

*Index Terms*—Mental Health, Depression Diagnosis/ Screening, Ensemble Learning, Explainable AI, Speech

## I. INTRODUCTION

Depression is a common mental health (MH) disorder with high prevalence worldwide affecting 3.8% of the population, including 5% of adults [1]. Psychological evaluation is the primary way to diagnose depression where a MH professional may ask questions about one's symptoms, thoughts, feelings, and behavior patterns. The American Psychiatric Association's Diagnostic Statistical Manual of Mental Disorders, Fifth Edition (DSM-5) is the handbook used by MH professionals in the United States and serves as an important resource worldwide [2]. Screening for depression is recommended in various medical settings (e.g., cardiovascular, perinatal, primary care) [3]. In the U.S., it is commonly conducted via the Personal Health Questionnaire depression scale (PHQ-8), an 8-item self-administered instrument with each item capturing different depression symptoms such as little interest in doing things or loss of appetite [4]. Despite the considerable clinical efforts on well-defined diagnostic criteria and screening tools, depression is misdiagnosed in 30-50% of female patients identifying them as depressed when they are not [5]. Additionally, primary care physicians recognize depression in only about half of

This work was supported by the National Science Foundation (CA-REER: Enabling Trustworthy Speech Technologies for Mental Health Care: From Speech Anonymization to Fair Human-centered Machine Intelligence, #2046118, PI: Chaspari). The code is available at https://github.com/HUBBS-Lab/speech-depression-ensemble-learning

the depressed patients they see [3]. This is largely due to the disorder's heterogeneity, overlapping diagnostic criteria, periodic changes in diagnostic categories, and high levels of somatization and differential reporting of medical symptoms.

Speech carries important information on depression [6]. Psychomotor symptoms associated with depression can be reflected in prosody [7], spectrotemporal characteristics [8], and vocal fold excitation [9]. Automated assessment systems, combining speech measures with machine learning (ML) algorithms, can potentially augment depression diagnosis and screening via offering clinicians data-driven insights that can complement their existing investigative techniques. However, depression identification accuracy is influenced by the trade-off between complexity and interpretability of the ML system. While simple systems relying on feature engineering are intuitive but often depict poor performance [10], deep learning systems, learning embeddings from raw speech, can achieve better results albeit being highly complex [11]. This underscores the need to design new explainable techniques for deep learning models to balance performance and interpretability.

Researchers have investigated ways to harness existing knowledge to improve ML explainability. By guiding the model on how to process information associated with the input or the output, it might learn in a human-understandable way and become more explainable. Prior work in informed machine learning (IML) [12] has proposed integrating context-specific knowledge that is formalized using representations such as equations, rules, or graphs. Evidence from speech production suggests that depression can influence the motor control and subsequently vowel generation [13], [14]. For instance, individuals with depression depict reduced vowel space, defined as the frequency range between the first and second vowel formant (i.e., F1 and F2), compared to their healthy counterparts [13]. Motivated by these observations, prior work proposed a vowel-dependent deep learning approach that learned depression-specific spectrotemporal patterns at the vowel-level [15], [16]. This proposed method outperformed approaches that model the spectrotemporal information in speech without considering vowel information, while the conducted explainability analysis indicated the importance of spectrotemporal patterns modeled at the vowel-level.

Ensemble learning aggregates multiple diverse models into a final system [17], a practice that can often lead to better results compared to using a single model due to the increase in the diversity of the system [18]. Ensemble learning can be explainable, as it tends to rely on simple base models that are easy to interpret individually, and the combination of these simpler models often follows an intuitive way. Because of its potential for explainability, ensemble learning has been proposed in clinical-related tasks. For example, Hu et al. explored various interpretable tree-based ensemble methods on early risk stratification of ischemic stroke and identified important factors contributing to model predictions [19]. Bouazizi et al. also predicted stroke using electroencephalogram signals with ensemble learning models integrated with global and local model explanation techniques [20]. Huo et al. proposed a Mixture of Experts (MoE) approach via a sparse gating network to predict patient risk prediction from electronic health record (EHR) data [21]. This can promote interepretability by breaking down the model into specialized, simpler components, providing transparency in routing decisions.

Here, we design an explainable ML algorithm for depression identification from speech. The explainability is integrated in two ways. First, inspired by recent successes in IML, we consider semantically meaningful linguistic units and use pre-trained vowel-based embeddings specifically for depression classification [16]. Second, we incorporate an ensemble learning approach that models high-level system decisions by breaking down the depression identification problem into its constituent parts, associated with the degree of depression severity or different depression symptoms. We examine a "bottom-up" ensemble learning approach consisting of 8 models, each predicting the score of an individual PHQ-8 survey question linked to a depression symptom. The total PHQ-8 score is aggregated based on the predictions of all models. By concentrating on individual PHQ-8 items, each model is able to distinguish specific depression symptoms. Additionally, the narrow range of prediction scores for each item (i.e., 0-4) depicts computational advantages compared to predicting the entire range of the overall PHQ-8 score (i.e., 0-24). We also investigated a "top-down" approach that relies on a MoE with a router module to identify the depression severity level (no, mild, moderate, moderately severe, severe). The router directs a sample to an expert model specialized in estimating a score within a specific severity level. Both proposed systems are designed to ensure that the classification and the regression results always align. In the bottom-up system this alignment is achieved because the binary and 5-class outcomes are directly derived from the predicted PHQ-8 score using established thresholds. Similarly, in the top-down system, the predicted PHQ-8 score is always within the range of the predicted depression class (e.g., none, mild, moderate, moderately severe, severe). Results indicate that both systems exhibit performance comparable to state-of-the-art when compared with several baselines [10], [16], [22]–[26] for both classification and regression tasks, highlighting their robustness. Also, the bottom-up system PHQ-8 estimates are

less influenced by the data mean/median value compared to prior work [10]. This reduces the likelihood of contradictory decisions between classification and regression, showcasing the consistency of our approach. Finally, we discuss explainability of the proposed methods and how these could be used by clinicians to augment depression diagnosis and screening.

## II. BACKGROUND & RELATED WORK

### A. The Patient Health Questionnaire-8 (PHQ-8)

The PHQ-8 scores individuals based on their responses to eight questions, each reflecting different domains of depressive symptoms [27]. These include interest or pleasure in activities, feelings of hopelessness or depression, sleep disturbances, energy levels, appetite or weight changes, feelings of failure or guilt, concentration difficulties, and psychomotor agitation or retardation. Each item is rated on a scale from 0 (not at all) to 3 (nearly every day), providing a cumulative score that helps to assess the overall severity of depressive symptoms. The total score ranges from 0 to 24 and relies on the following cutoff points for assessing the severity of depression: 0-4 (none, minimal), 5-9 (mild), 10-14 (moderate), 15-19 (moderately severe), 20-27 (severe). A single cutoff of 10 is also used to identify major depression.

A common practice is to interpret the aggregate PHQ-8 scores without understanding the contribution of the individual items, which may have varying degrees of clinical importance [28]. For example, item 2 of the PHQ-8 (i.e., "Feeling down, depressed, or hopeless") reflects a more severe symptom compared to item 7 ("Trouble concentrating on things"). However, the PHQ-8 scoring algorithm treats both items equally when computing the total score. In addition, tracking the individual PHQ-8 items may help clinicians better interpret symptoms and identify the efficacy of treatment strategies. Training separate ML models for each individual PHQ-8 score can contribute to indicating the likelihood of depression based on different symptoms. This can increase flexibility in decision-making, since clinicians can weigh the importance of different aspects of the questionnaire based on the individual's profile and associate types of behaviors with each item.

### B. Ensemble Learning for Depression Identification

Previous research has applied ensemble learning methods on the task of depression identification and treatment prediction. Nguyen et al. designed a deep stacked ensemble system, DeSGEL, to identify depression using accelerometry data from wearable devices [29]. Ansari et al. trained ensemble text-based classifiers on online social content to detect depression [30]. Vazquez et al. trained 50 1-dimensional convolutional neural networks (CNN) with different initializations based on a sequence of log-spectrograms and applied an ensemble averaging algorithm to provide a depression decision per speaker [31]. Aharonson at al. also designed two types of ensemble methods aiming to first use a classifier to classify depression levels (i.e., binary or multi-class) based on speech, followed by training a regression model to estimate the exact PHQ-8 score within each predicted depression severity class [32]. Evaluation of the system performance was conducted via randomly splitting the data into 70/30% train/test set, rather

than in the speaker-independent manner commonly used in prior work [10], [11], [22]. In regard to treatment prediction, Pei et al. used the ensemble of SVM models to decide the usefulness of a treatment by measuring the early-response to treatment via biomarkers [33]. Pearson et al. combined a random forest and an elastic net to predict depressive symptoms to forecast the effectiveness of an internet-based intervention using information such as one's psychopathology, demographics, treatment expectancies, and usage [34].

*C. Depression Identification from Speech*

Beyond ensemble learning for speech-based depression identification, prior work has explored various approaches. The DAIC-WOZ dataset is a common evaluation benchmark extensively used in prior work [35]. Chen et al. proposed a hierarchical self-attention structure, called "SpeechFormer," that considers the structural characteristics of speech learning acoustic embeddings at the frame, phoneme, and word-level. These are merged at the utterance-level to generate a global representation for a subsequent classification or regression task. SpeechFormer was trained on DAIC-WOZ resulting in a macro-F1 score of 0.694 for the binary classification task [11]. Du et al. leveraged linear predictive coding (LPC) and Mel-frequency cepstral coefficients (MFCC) features to simulate the processes of speech production and perception, respectively yielding a macro-F1 score of 0.75 [25]. Feng & Chaspari designed a vowel-based approach to extract and utilize the vowel information in speech with a 0.73 macro-F1 score [16]. Wang et al. obtained a similar score while preserving speaker privacy using a speaker disentanglement method that adds a speaker identification loss in an adversarial manner to the depression detection loss [23].

Beyond depression classification, researchers have also designed regression methods for estimating the PHQ-8 score. Fang et al. designed a multi-level attention mechanism that learns audiovisual and text embeddings, followed by fusing those with an attention fusion network, resulting in a 6.13 root mean square error (RMSE) [24]. Lin et al. used a 1D CNN over the speech spectrogram to model interactions within the frequency bands, followed by a bidirectional long short-term memory (BiLSTM) neural network [22]. This resulted in a 4.25 mean absolute error (MAE) and 5.45 RMSE. Grounded in the hypothesis that speech representations of personal identity (i.e., speaker embeddings) can improve depression identification, Dumpala et al. extracted speaker embeddings from models pre-trained on speaker identification using a large speaker sample from the general population without depression information [26]. When combined with acoustic features widely used for depression, such as the OpenSMILE ones, these speaker embeddings resulted in a 0.66 macro-F1 score and a 6.01 RMSE. However, the tasks of depression classification and regression were considered separately.

*D. Limitations of Prior Work and Study Contributions*

Despite the promising results, previous studies depict the following limitations. First, these studies often prioritize minimizing the MAE, resulting in predictions that tend to converge towards the mean/median PHQ-8 of the training set [36].

Simple regression models, such as random forests [34], are particularly prone to this effect. Methods more sophisticated than random forests, such as the ones proposed by Lin et al. [22], are less affected but still struggle to cover the entire range of PHQ-8 scores. When these scores are assigned to a depression severity group via thresholding (e.g., 10 as the binary threshold between depression and non-depression [27]), this often results in wrong decisions. Second, previous research often considers the depression classification and regression tasks separately. This approach introduces additional challenges associated with discrepancies between potentially different decisions resulting from the classification and regression models. For example, a binary classifier might classify a patient in the depression category, whereas the regressor could predict a PHQ-8 score below 10, thus conflicting with the classifier's outcome. This "disconnect" raises concerns about system interpretability potentially hindering the user's trust. Third, prior work has focused on the estimation of the final PHQ-8 score without considering each separate PHQ-8 item. Learning item-specific models can potentially increase model flexibility, since different input parts or modalities might contribute differently to an individual item. This can also increase the explainability of automated methods allowing clinicians to link specific patient behaviors to particular symptoms.

The contributions of this paper are as follows: (1) While prior work has focused on depression identification using the aggregate PHQ-8 score, this paper examines ML systems that estimate each PHQ-8 item separately (bottom-up system). Since PHQ-8 items represent various symptoms, this can enhance system interpretability via allowing different spectrotemporal speech characteristics to be associated with each item; (2) In contrast to prior research where depression severity classification and estimation were performed separately, this study leverages ensemble learning approaches that conduct both tasks jointly promoting the alignment between regression outcomes (i.e., specific PHQ-8 scores) and classification outcomes (e.g., levels of depression) (top-down/bottom-up systems); and (3) The proposed explainable bottom-up and top-down systems achieve performance comparable to state-of-the-art (SOTA) in DAIC-WOZ, while they enhance the interpretability of speech-based depression identification, which is uncommon in SOTA baselines (e.g., [11], [23]).

## III. Data Description

Data come from the DAIC-WOZ dataset [35] that includes 142 clinical interviews divided into training (107 participants), development (35 participants), and testing (labels withheld from the public) [37]. The PHQ-8 score is used as a depression measure. Using 10 as a threshold in the binary classification [38], we have 77 healthy and 30 participants with depression for training, and 23 healthy and 12 participants with depression in the development set. Following common cutoff points (5, 10, 15, 20) for the 5-way classification [27], there are 47, 29, 20, 7, and 4 participants with no, mild, moderate, moderately severe, and severe depression, respectively, in the training set, and 17, 6, 5, 6, and 1 participants with no, mild, moderate, moderately severe, and severe depression, respectively, in

the development set. Since the labels of the testing are not available, the development set is used as testing in this paper.

## IV. PROPOSED METHOD

Here we describe the process of extracting vowel-based speech embeddings (Section IV-A), followed by data augmentation to tackle the observed class imbalance (Section IV-B). The proposed bottom-up and top-down ensemble systems are described in Sections IV-C and IV-D, respectively.

### A. Vowel-Based Speech Embeddings

We employ an explainable open-source [1] pre-trained encoder system [16] that depicts binary classification performance similar to SOTA in DAIC-WOZ. The system relies on evidence from speech production indicating that depression can influence vowel generation [13], [14], thus its training extracts vowel-based embeddings learned from a vowel classification module. This contributes to system explainability via allowing users to observe depression-specific spectrotemporal variations at the vowel-level. The system takes at the input speech spectrograms from a group of utterances and processes them through a series of convolutional layers with a spatial pyramid pooling (SPP) layer that conducts vowel classification. The log-Mel spectrogram is extracted for every 250ms speech segment using a 512-sample Fast Fourier Transform (FFT) window length, 128-sample hop length, and 128 Mel bands, resulting in a spectrogram patch size of $(128, 28)$. The SPP learns short-term spectrotemporal information throughout an utterance and produces a sequence of embeddings that are used as the input for a fully-connected layer that outputs a binary decision for the depression classification task for the considered group of utterances [16]. We extract the output from the intermediate layer of the system located right before the final output layer, generating a 64-dimensional vowel-based embedding $\mathbf{v_i}$ for each utterance group $i$ of a speaker.

### B. Data Augmentation

We tackle the issue of class imbalance commonly encountered in depression identification. First, we oversample utterance groups from the minority classes such as moderate and severe depression. The ratio of over-sampling is based on the class distribution in the training set ensuring approximately equal representation across classes. Additionally, we apply a data augmentation method by [16], [39]. This method randomly perturbs a subset of utterances in a group, while preserving from perturbation the utterances that are deemed as the most salient in relation to the depression outcome. We adhere to the augmentation settings in [16], [39], perturbing 6 utterances and preserving the most salient 21 utterances.

### C. Bottom-up Ensemble System Design

The first ensemble system models a bottom-up approach that learns a regression/classification task for each PHQ-8 score, followed by aggregating the decisions from the individual models to derive the final PHQ-8 score (Figure 1). We first conduct over-sampling and data augmentation (Section IV-B) and extract the vowel-based embeddings $\{\mathbf{v_i}\}_i$ of the augmented dataset (Section IV-A). For each PHQ-8 item $k$ ($k = 1, \ldots, 8$), we apply a transformation $f_k$ to the embedding in

[1]https://github.com/HUBBS-Lab/ICASSP-2023-Augmented-Knowledge-Driven-Speech-Based-Method-of-Depression-Detection

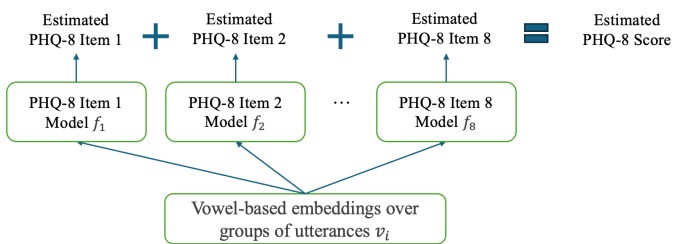

Fig. 1: A schematic illustration of the bottom-up system. Separate PHQ-8 items are estimated based on individual models and are further aggregated into the final PHQ-8 score.

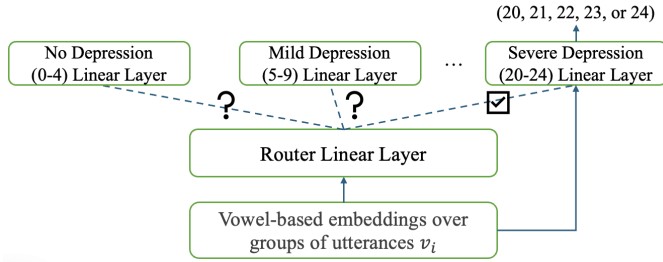

Fig. 2: A schematic illustration of the top-down system. Each expert is trained on a depression severity class separately. The router selects an expert (e.g., severe depression in the example) that predicts a score within the corresponding range (e.g., 20 to 24 for severe depression).

order to obtain a decision $f_k(\mathbf{v_i}) \in 0, \ldots, 3$ that represents the severity score of the corresponding item. A softmax activation function is applied due to its ease of implementation and compatibility with the existing encoder system (Section IV-A). This further promotes the explainability of the overall system. The system parameters are learned using cross-entropy loss. The oversampling ratio for each PHQ-8 item varies based on the class distribution within the respective training set, while the rest of the hyper-parameters (i.e., Adam optimizer, 0.001 learning rate, 5 epochs) remain the same across all models.

During testing, samples from each speaker belonging to the test set are segmented into non-overlapping utterance groups, and their embeddings $\mathbf{v_i}$ for each utterance group $i$ are obtained using the encoder (Section IV-A). To estimate each PHQ-8 item for a test speaker, we aggregate predictions $f_k(\mathbf{v_i})$ from all utterance groups and use the mode as the final prediction for each item $k$, such that $g_k = mode_i (f_k(\mathbf{v_i}))$. The mode is selected as it is less prone to outliers. To obtain the overall estimated PHQ-8 score $h$ for the test speaker, we sum all predictions from the eight individual PHQ-8 scores such as $h = \sum_{k=1}^{8} g_k$. The corresponding binary outcomes are derived directly from the predicted PHQ-8 score $h$ using the threshold of 10 (Section II-A), while predictions for the 5-class outcome are similarly derived using the corresponding thresholds (i.e., 5, 10, 15, and 20; Section II-A).

### D. Top-down Ensemble System Design

The second ensemble system relies on a MoE including a router and five experts, where each experts correspond to a depression severity category (i.e., none, minimal, mild, moderate, moderately severe). The router determines which

expert to employ for a given sample [40] (Figure 2). This approach is widely used in training large language models (LLM), to reduce the overall number of parameters needed [41]. Here, we limit the number of experts to five to maintain system explainability. Each expert is assigned to a depression severity level, including none, mild, moderate, moderately severe, and severe (Section II-A). The router selects one of these experts to take a final decision for an input.

Similar to the bottom-up approach (Section IV-C), we used an over-sampling and data augmentation (Section IV-B) and extracted the vowel-based embedding (Section IV-A), $\{\mathbf{v_i}\}_i$ for all utterance groups of the augmented dataset. Following that, the router $r$ took as an input the vowel-based embedding and selected one out of the five experts such that $r(\mathbf{v_i}) \in \{1, \ldots, 5\}$, where $r$ is expressed via a linear transformation layer that outputs one of the five depression categories. To determine which expert to utilize for a test speaker, we obtain predictions from the router for each utterance group and select the expert via the mode, such that $d = mode_i(r(\mathbf{v_i}))$. Once the expert $d$ is chosen, we input the vowel-based embeddings to the selected expert $s_d(\mathbf{v_i})$ and use soft voting to determine the final PHQ-8 score such that $z = argmax_i(s_d(\mathbf{v_i}))$. Similar to the bottom-up design, both the router $r$ and each expert $s_j$, where $j = 1 \ldots, 5$, are implemented as linear transformations with a softmax activation function and cross-entropy loss. Given the clear learning objectives of each module, we train the router and experts separately, which is different from the simultaneous training approach used in LLMs [41]. The router is trained on the entire training set, while each expert is trained solely on the subset of the training set that corresponds to its assigned depression severity level. Other training parameters include an Adam optimizer (learning rate 0.001) and 10 epochs. Binary and 5-class predictions are derived using the same method as in the bottom-up approach.

## V. RESULTS

### A. Depression Identification Performance

Evaluation of the performance of the proposed systems is conducted via the macro-F1 score for the binary and 5-way depression classification tasks, and MAE, RMSE, and Pearson's correlation between actual and estimated PHQ-8 score for the regression task. These are reported at the speaker-level based on the speakers included in the validation set of DAIC-WOZ. The proposed systems are compared against baselines with SOTA results in speech-based depression identification, including methods that leverage spectrogram-based speech embeddings [16], [22], [23], prosodic, spectrotemporal, and speech production measures [10], [24], [25], and their combination [26] (see Section II-C for a detailed description).

We report the performance of the proposed and baseline systems in Table I. Given the complexity of many of the baseline systems, the majority of presented performance metrics are based on the results reported in the corresponding papers. Since not all baseline systems conducted both classification and regression, some of the results are missing. The proposed ensemble learning systems are competitive in both the classification and the regression depicting the highest F1-scores in

TABLE I: Model performance of proposed and baseline methods. Macro-F1 score reported for binary and 5-way depression classification. Mean absolute error (MAE), root mean square error (RMSE), and Pearson's correlation were reported for depression severity estimation via regression. The best result per category is denoted with bold font.

| Method | Classification (Macro F1) | | Regression | | |
|---|---|---|---|---|---|
| | Binary | 5-way | MAE | RMSE | Correlation |
| AVEC (2017) [10] | 0.55 | - | 5.35 | 6.48 | - |
| Ensemble CNN (2020) [31] | 0.73 | - | - | - | - |
| BiLSTM/1D CNN (2020) [22] | 0.55 | - | **4.25** | **5.45** | - |
| SpeechFormer (2022) [11] | 0.694 | - | - | - | - |
| Speaker Disentangle (2023) [23] | 0.735 | - | - | - | - |
| Multi-level Attention (2023) [24] | - | - | 5.21 | 6.13 | - |
| Speaker Embedding (2023) [26] | 0.66 | - | - | 6.01 | - |
| Proposed Top-down Ensemble | 0.755 | 0.246 | 4.66 | 5.77 | **0.49**[*] |
| Proposed Bottom-up Ensemble | **0.762** | **0.338** | 4.91 | 6.89 | 0.36[*] |

[*]: $p < 0.05$

classification and the second lowest MAE and RMSE scores in regression. The bottom-up system has better performance compared to the top-down system. In comparison to the baseline model that employed the vowel-based embeddings [16], the proposed system yields a slight increase in classification performance and a decrease in the MAE score, while also offering the additional capability of the 5-way classification of depression severity. Despite the fact that better results in terms of regression were obtained by [22], the model in [22] is not explainable since it relies on spectrogram representations that do not take the hierarchical representation of speech and depicts lower F1-score compared to our methods.

### B. Comparison between Actual and Estimated PHQ-8

We further plot the estimated and actual PHQ-8 score for each test participant (Figure 3), as estimated by the proposed ensemble systems, the random forest regressor baseline [10] and BiLSTM/1D CNN baseline [22].

The number of testing samples in each category, from no to severe depression, is 17, 6, 5, 6, and 1. The bottom-up system has F1 scores of 0.65, 0.40, 0.44, 0.20, and 0.0 (macro-F1 = 0.338) for each category. The top-down system has F1 scores of 0.56, 0.32, 0.35, 0.0, and 0.0 (macro-F1 = 0.246). Visual inspection indicates that the random forest regression [10] yields PHQ-8 estimates with the least fluctuations, making it less reliable for extreme classes. The proposed top-down system and the baseline by [22] still yield PHQ-8 estimates close to the median sample value but with more fluctuations compared to [10]. However, the top-down system overall had better performance compared to [22] (Table I). The top-down system falls short in the 5-way classification task, particularly for scores with high PHQ-8 values, indicating its limitation in identifying extreme cases. The bottom-up system demonstrates the highest fluctuation in predictions, covers the entire range of PHQ-8 scores, and appears to classify participants with moderate to severe depression. One possible explanation is that the bottom-up system learns a smaller range (i.e, 0-3) for individual PHQ-8 items separately, rather than the larger range (i.e, 0-24) of the aggregate PHQ-8 score.

### C. Estimation of PHQ-8 Items in Bottom-up Approach

We compare the actual and estimated PHQ-8 items to understand the effectiveness of the proposed bottom-up system in

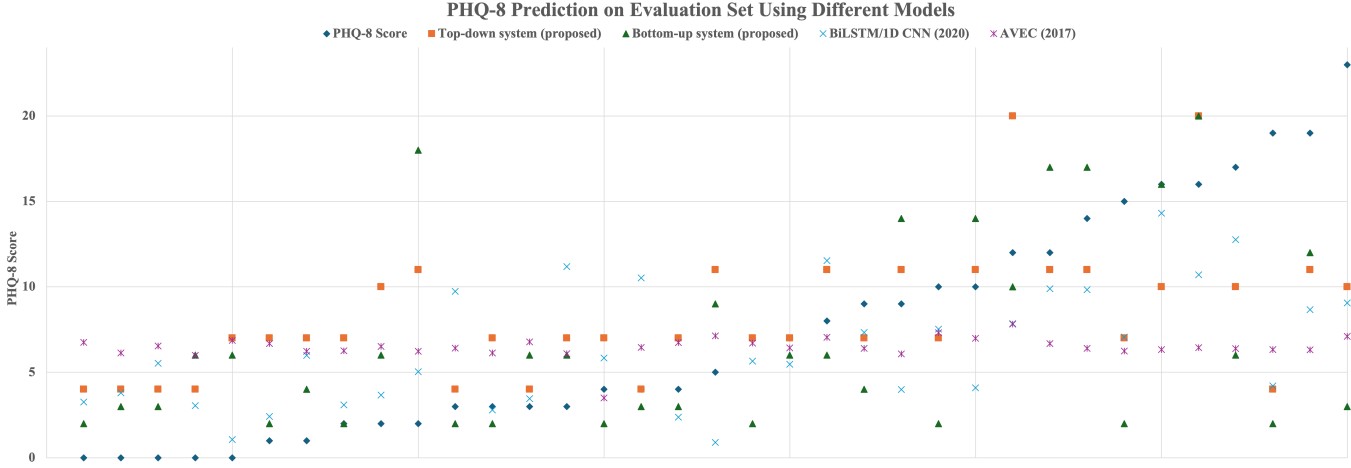

Fig. 3: 2D scatter plot between actual and predicted PHQ-8 score from different systems. The x-axis depicts the participants of the test sample ordered in ascending order of the actual PHQ-8 score.

TABLE II: Pearson's correlation, mean absolute error (MAE), and root mean square error (RMSE) between true and predicted individual PHQ-8 items. Results with correlations larger than $0.2$ are highlighted with bold text.

| PHQ-8 Score Index | Correlation | MAE | RMSE |
|---|---|---|---|
| Q1 No Interest (Little interest or pleasure in doing things) | 0.108 | 1.314 | 1.724 |
| Q2 Depressed (Feeling down, depressed, or hopeless) | 0.025 | 1.0 | 1.384 |
| **Q3 Sleep (Trouble falling or staying asleep, or sleeping too much)** | **$0.297^{\dagger}$** | **1.029** | **1.414** |
| Q4 Tired (Feeling tired or having little energy) | -0.212 | 0.857 | 1.146 |
| **Q5 Appetite (Poor appetite or overeating)** | **0.252** | **1.0** | **1.444** |
| **Q6 Failure (Feeling bad about yourself, or that you are a failure, or have let yourself or your family down)** | **$0.298^{\dagger}$** | **0.914** | **1.434** |
| Q7 Concentrating (Trouble concentrating on things, such as reading the newspaper or watching television) | 0.112 | 0.971 | 1.146 |
| **Q8 Moving (Moving or speaking so slowly that other people could have noticed. Or the opposite – being so fidgety or restless that you have been moving around a lot more than usual )** | **0.246** | **0.629** | **1.014** |

$\dagger$: $p < 0.1$

estimating each score separately. For this purpose, we compute the Pearson's correlation, MAE, and RMSE between actual and predicted values of each PHQ-8 score. The proposed bottom-up system depicts a Pearson's correlation larger than 0.2 for four out of the eight PHQ-8 scores, including 'sleep,' 'appetite,' 'failure,' and 'moving' (Table II). In contrast, we observe lower Pearson's correlations for PHQ-8 items such as Q1: no interest, Q2: depressed, Q4: tired, and Q7: concentrating. A potential reason might be the sub-items are too broad to predict using speech patterns. Also for the value 3 of the Q4 score had a much higher ratio in the development set (20%) compared to the same ratio in the training set (8.4%).

*D. Internal Consistency in Bottom-Up Approach*

The internal consistency among the items of a survey is often used for survey evaluation assessing correlations between different survey items that are meant to measure the same construct [42]. Here, we compute the internal consistency among the actual scores of the PHQ-8 items in the data, and the scores estimated by the bottom-up system. We anticipate that the level of internal consistency should be maintained between the actual and estimated scores. We use Cronbach's alpha, $\alpha$, which is widely used for this purpose [43]. The actual scores of the PHQ-8 in the development set of our data have $\alpha_1 = 0.910$, which is close to the same metric reported in other datasets (i.e., $\alpha_2 = 0.82$ [44]). The scores estimated

by the bottom-up stsrem have $\alpha_3 = 0.886$, which is close to the one from the actual scores $\alpha_1$. This suggests that the level of internal consistency is maintained between the actual and estimated PHQ-8 items.

*E. Correlation Between Estimated PHQ-8 Score and Acoustic Measures in Top-Down Approach*

We investigate the explainability of the top-down system by computing the Pearson's correlation between predicted PHQ-8 scores and several prosodic measures commonly used for depression identification [6], including speech percentage, mean fundamental frequency (F0), standard deviation of F0, jitter, shimmer, and loudness. We found a significant association between estimated PHQ-8 and speech percentage ($r = -0.160$, $p = 0.05$), F0 standard deviation ($r = 0.223$, $p < 0.01$), and jitter ($r = -0.160$, $p < 0.01$) indicating that the proposed system captures interpretable acoustic factors associated with the focal clinical outcome [45]. Mean F0 also depicted a significant association with the PHQ-8 score ($r = 0.38$, $p < 0.01$). Although prior work has reported mixed findings regarding mean F0 as a robust marker of depression [46], our result might suggest that the model may capture gender-related information that is closely related to F0 [47].

VI. DISCUSSION

This study examined two ensemble learning systems, a bottom-up and a top-down approach, for the identification of

depression from speech. The bottom-up system, comprising of eight models that independently score each item of the PHQ-8 questionnaire before aggregating them into the final PHQ-8 score, achieves F1-scores of 76.2% and 33.8% for binary and 5-way depression classification tasks, respectively, outperforming all considered baselines (Table I). The top-down system, employing a MoE method directing input to a specific model for depression severity via a router, demonstrates the second-best classification results and comparable performance to [25], which utilizes speech production features. Both systems exhibit the second and third lowest MAE, respectively, being only lower than the BiLSTM/1D CNN [22], which notably achieves a substantially lower F1-score (i.e., 55%). Compared to models also utilizing spectrogram embeddings [11], [16], [26], both systems demonstrate superior performance, indicating the efficacy of ensemble learning approaches in the focal task. Moreover, the bottom-up approach yields a well-balanced distribution of estimated PHQ-8 scores across various depression severity levels (Figure 3), a challenge for many other systems. Our AI systems could enhance diagnostic accuracy by providing detailed and objective insights into patient speech patterns related to depression, complementing traditional assessment methods. Furthermore, they could be integrated with existing clinical workflows, potentially streamlining assessments and saving clinicians' time.

In addition to the effectiveness in reliably identifying depression, the proposed models offer several explainability advantages. Through estimating individual scores for each PHQ-8 item, the bottom-up system allows health professionals to interpret the AI model decisions in association to specific symptoms. During interviews conducted by the authors with clinicians, one emphasized the importance of associating explainable systems with specific diagnostic criteria, stating: "*If there were more details on what these utterances are picking up as far as like the diagnostic criteria, then I might feel a little bit more comfortable.*" This suggests that the proposed bottom-up system could potentially serve as an assistive tool for helping clinicians understand specific depression symptoms, which could be further explored through targeted follow-up questions focused on symptoms identified by the model. The PHQ-8 items estimated by the bottom-up system further demonstrate internal consistency similar to the actual scores. This can contribute to system explainability by ensuring that the estimated scores reflect the same relative relationships as the actual PHQ-8 items, helping the system avoid producing conflicting results for associated items. Through the bottom-up system, we can further identify items that are challenging to estimated (e.g., Q2, Q4; Table II). This can potentially help in determining system bottlenecks. As part of our future work, we will investigate usability, explainability, and human trust via user studies involving healthcare professionals.

The top-down system can also achieve explainability in the following ways. Since each expert model is specialized in identifying different levels of depression severity, the association between the learned embeddings and the outcomes for each model may reveal spectrotemporal patterns specific to each severity level. This can inform healthcare experts about specific speech patterns indicative of depression severity to observe. In addition, the router determines which expert model to use, offering further insight into the model's decision-making process, including the features influencing expert selection. While this paper employs a simple router, our future work will explore more advanced ones, such as gating networks and hierarchical routers to better capture the hierarchical nature of the focal clinical outcome.

Despite the encouraging results, the proposed systems depict the following limitations. The proposed approach relies solely on acoustic features and overlooks valuable linguistic content. While combining acoustic and linguistic features into an explainable system poses a challenge due to their distinct temporal granularities, it can provide deeper insights into a one's cognitive and emotional state potentially enhancing the accuracy of depression identification. Additional contextual variables such as information on one's demographics, medical history, home environment, and lifestyle can further be incorporated into the system to improve automatic diagnosis. Additionally, we evaluated the proposed system on a single subset of DAIC-WOZ, a common practice in previous studies [10], [11], [15], [22], [23]. However, the external validity of the proposed systems on other datasets has not yet been established. Finally, the proposed systems were developed specifically for the English language, focusing on vowel-based information found in American English. Consequently, additional work is needed to generalize the systems to other languages, considering the linguistic and phonetic variations that influence depression markers across different languages.

## VII. CONCLUSIONS

We examined two types of ensemble learning systems for speech-based depression classification and depression severity estimation. The first type relies on a bottom-up approach that estimates separate items of the PHQ-8 score before merging those into an aggregated score. The second type relies on a top-down MoE system that uses model experts specifically trained to identify nuances within each depression severity level. Both systems achieved results comparable to SOTA when compared with a diverse set of baselines that take into account commonly used speech features related to prosody, spectrotemporal variations and speech production, as well as acoustic embeddings trained on raw speech spectrograms. In addition to the observed performance advantages, the bottom-up design yields PHQ-8 estimates that cover the entire range of the score, potentially being able to reliably detect extreme cases. The estimated PHQ-8 items resulting from the bottom-up system further depict internal consistency comparable to the actual scores, suggesting that this system can maintain relative associations within the PHQ-8 items. The proposed ensemble learning methods, particularly the bottom-up design, depict additional explainability advantages since they can break down the problem of depression identification to individual symptoms and inform healthcare experts on specific speech patterns to pay attention to for a given depression severity level.

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
