# OpenReview forum: "Robust and Explainable Depression Identification from Speech Using Vowel-Based Ensemble Learning Approaches"
_IEEE.org/EMBS/BHI/2024/Conference — IEEE BHI'24_

### Official Review · Reviewer_uUdG · 2024-07-29
**Robust and Explainable Depression Identification from Speech Using Vowel-Based Ensemble Learning Approaches**

**Overall Rating:** 6
**Confidence:** 1

**Other Quality Metrics:**

(a) clarity of writing: good
(b) clinical significance: good
(c) Methodological Novelty: fair
(d):Experiments and Results:fair

**Questions For The Authors:**

1. What specific aspects of depression severity does your model address?
2.The model evaluates individual symptoms associated with the PHQ-8 items, enhancing interpretability by linking speech characteristics to specific depression symptoms. How does the proposed bottom-up system differ from traditional methods in measuring depression?
3.The bottom-up system estimates each PHQ-8 item separately, allowing for a more nuanced understanding of symptoms rather than relying on an aggregate score, which is common in prior research.
What measures have you taken to ensure the explainability of your AI model?
4.The model provides individual scores for each PHQ-8 item and allows healthcare professionals to interpret decisions based on specific symptoms, enhancing user trust and understanding.
How did you address class imbalance in your dataset?
5.The study employed oversampling of minority classes and data augmentation techniques to ensure equal representation across different depression severity levels.
What are the potential implications for healthcare professionals using your system?
6.The system could assist clinicians in diagnosing and understanding specific depression symptoms, thereby improving the quality of patient care.
7.How would clarification on the explainability of your model change its score?
8.A better understanding of the model’s explainability may enhance its perceived reliability, potentially leading to a higher score due to increased confidence from evaluators.
Would more detailed information on symptom-specific identification affect your evaluation?
9.Can you elaborate on how your model maintains consistency in PHQ-8 score estimation?
10.Understanding the internal consistency of estimated scores compared to actual scores may impact scoring, as it reflects the robustness of the model in clinical applications.
What challenges did you encounter while estimating specific PHQ-8 items?
11.Insights into challenges with certain items (e.g., Q2, Q4) may reveal limitations of the model, affecting its overall scoring by highlighting areas for improvement or further research.
How might the results of user studies involving healthcare professionals shape your research further?

**Strengths:**

-The authors succeed to hide their background and names even the available scripts on Github were uploaded with anonymous account.
-The draft  the IEEE BHI format writing conditions.
- the topic fits the BHI conference context.
-This paper presents several notable strengths that contribute to its significance in the field of depression identification through speech analysis. The promising aspects include((page 3,6 and 7):
°Innovative Framework: The paper introduces "SpeechFormer," a hierarchical self-attention structure that effectively integrates acoustic embeddings at multiple levels (frame, phoneme, word) to enhance classification and regression tasks related to depression detection.
°Enhanced Performance Metrics: The proposed methods achieve macro-F1 scores comparable to state-of-the-art systems, demonstrating a macro-F1 score of 0.694 for binary classification and 0.75 using linear predictive coding features, indicating robust performance in identifying depression.
°Joint Task Approach: Unlike prior studies that addressed depression severity classification and estimation separately, this paper employs ensemble learning methods to conduct both tasks simultaneously. This approach improves the alignment between the outputs of regression (specific PHQ-8 scores) and classification (levels of depression).
°Interpretability Improvement: The paper focuses on estimating each item of the PHQ-8 score separately, which enhances system interpretability. This allows clinicians to link specific speech characteristics to individual symptoms of depression.
°Explainable Systems: The integration of explainability in both the bottom-up and top-down systems allows for better understanding of how speech features correlate with depression symptoms, making the system more user-friendly for healthcare professionals.
°Use of Comprehensive Dataset: Utilizing the DAIC-WOZ dataset, which includes a substantial number of clinical interviews, provides a solid foundation for training and validating the proposed models, enhancing the reliability of results.
°Focus on Clinical Relevance: The study's considerations for clinical implications, such as assisting clinicians in identifying specific depression symptoms, highlight its potential to influence practical applications in mental health care.
°Addressing Limitations of Prior Work: The paper critically discusses the limitations of previous methods, particularly their tendency to converge towards mean values in predictions, and proposes solutions to these challenges through more sophisticated models.
°Future Research Directions: The authors outline future work that will explore usability, explainability, and trust in the system through user studies, indicating a commitment to ongoing improvement and adaptation to clinical needs.
°Contribution to Field: By combining advanced machine learning techniques with a focus on explainability and clinical applicability, this paper contributes significantly to the development of tools for automated depression detection, setting a precedent for future research in this area.
The findings suggest that specific speech features, such as prosody and vocal fold excitation, are significant markers for depression, providing actionable insights for healthcare professionals.

**Summary Of The Paper:**

-The context of the study focuses on the application of machine learning algorithms to identify depression through the analysis of speech, particularly using vowel-based embeddings and ensemble learning techniques.
-The research highlights how these methods perform comparably to leading benchmarks and emphasizes their potential to provide explainable insights that could aid clinicians in diagnosing and screening for depression.
-This is particularly significant given the challenges in accurately diagnosing depression, which is prevalent globally and often misdiagnosed due to its varied presentation and overlapping criteria.

**Weaknesses:**

- Class Imbalance: The training set had a notable class imbalance, with significantly fewer participants diagnosed with severe depression compared to those classified as healthy or mildly depressed. This could affect the model's ability to generalize effectively across all severity levels of depression.
- Lack of External Validation: The model's performance was evaluated using the development set, which might not provide an adequate assessment of its effectiveness in real-world applications. External validation with independent datasets would strengthen the findings.
- Complexity of Models: The proposed ensemble learning methods, while innovative, may be overly complex, leading to challenges in practical implementation and interpretability for clinicians.
- Although the paper emphasizes explainability, the methods for linking specific patient behaviors to depression symptoms could be more clearly articulated, which may limit clinician confidence in using the tool.
-The reliance on speech-based features might overlook other important behavioral or contextual factors that contribute to depression, limiting the comprehensiveness of the assessment.

---

### Official Review · Reviewer_F3TH · 2024-08-09
**This study systematically demonstrates the validity and improvement of the proposed framework, leading to a better understanding of the addressed issue.**

**Overall Rating:** 7
**Confidence:** 3

**Other Quality Metrics:**

(a) Clarity of writing: good
(b) Clinical Significance: good
(c) Methodological Novelty: excellent
(d) Experiments and Results: good

**Questions For The Authors:**

Regarding Table 1, how does the performance compare to studies [15], [16], and [32]?

It is not clear exactly how the router is trained and how the training data optimizes the pre-trained encoder system from [16].

**Strengths:**

The major novelty of the proposed ensemble learning is its ability to classify depression symptoms and regress their severity levels jointly. This promotes the alignment of classification and regression outcomes, leading to better diagnosis of depression in clinical practice.

This study demonstrates its capability by conducting a thorough assessment of classification and regression analysis.

**Summary Of The Paper:**

This study attempts not only to improve the performance of identifing depression symptom and its severity level from speech, but also provide explainable insights to help clinicians in depression diagnosis. Towards this, the authors proposes vowel=based ensemble learning approachs in terms of buttom-up and top-down strategies. The proposed ensemble learning can lead to superior performance with respect to most adopted benchmark methods.

**Weaknesses:**

The authors are encouraged to perform a leave-one-subject-out validation procedure. This allows for the formation of the distribution of a performance metric. Table 1 can then assess the statistical difference between different methods. Additionally, the performance based on such a validation procedure infers better to a real patient scenario.

This study initially argues the importance of aligning classification and regression tasks. However, no relevant assessment or discussion is provided.

While presenting the difference between actual and predicted PHQ-8 in Fig. 3, categorizing the outcomes in terms of five severity levels may provide more insights into the proposed framework, particularly the interplay between sample size and performance per level.

---

### Decision · Program_Chairs · 2024-09-23

Accept